# MEMe: An Accurate Maximum Entropy Method for Efficient Approximations in Large-Scale Machine Learning

**DOI:** 10.3390/e21060551

**Published:** 2019-05-31

**Authors:** Diego Granziol, Binxin Ru, Stefan Zohren, Xiaowen Dong, Michael Osborne, Stephen Roberts

**Affiliations:** 1Machine Learning Research Group, University of Oxford, Walton Well Rd, Oxford OX2 6ED, UK; 2Oxford-Man Institute of Quantitative Finance, Walton Well Rd, Oxford OX2 6ED, UK

**Keywords:** maximum entropy, log determinant estimation, Bayesian optimisation

## Abstract

Efficient approximation lies at the heart of large-scale machine learning problems. In this paper, we propose a novel, robust maximum entropy algorithm, which is capable of dealing with hundreds of moments and allows for computationally efficient approximations. We showcase the usefulness of the proposed method, its equivalence to constrained Bayesian variational inference and demonstrate its superiority over existing approaches in two applications, namely, fast log determinant estimation and information-theoretic Bayesian optimisation.

## 1. Introduction

Algorithmic scalability is an important component of modern machine learning. Making high quality inference on large, feature rich datasets under a constrained computational budget is arguably the primary goal of the learning community. This, however, comes with significant challenges. On the one hand, the exact computation of linear algebraic quantities may be prohibitively expensive, such as that of the log determinant. On the other hand, an analytic expression for the quantity of interest may not exist at all, such as the case for the entropy of a Gaussian mixture model, and approximate methods are often both inefficient and inaccurate. These highlight the need for efficient approximations especially in solving large-scale machine learning problems.

In this paper, to address this challenge, we propose a novel, robust maximum entropy algorithm, stable for a large number of moments, surpassing the limit of previous maximum entropy algorithms [1,2,3]. We show that the ability to handle more moment information, which can be calculated cheaply either analytically or with the use of stochastic trace estimation, leads to significantly enhanced performance. We showcase the effectiveness of the proposed algorithm by applying it to log determinant estimation [4,5,6] and entropy term approximation in the information-theoretic Bayesian optimisation [7,8,9]. Specifically, we reformulate the log determinant estimation into an eigenvalue spectral estimation problem so that we can estimate the log determinant of a symmetric positive definite matrix via computing the maximum entropy spectral density of its eigenvalues. Similarly, we learn the maximum entropy spectral density for the Gaussian mixture and then approximate the entropy of the Gaussian mixture via the entropy of the maximum entropy spectral density, which provides an analytic upper bound. Furthermore, in developing our algorithm, we establish equivalence between maximum entropy methods and constrained Bayesian variational inference [10].

The main contributions of the paper are as follows:We propose a maximum entropy algorithm, which is stable and consistent for hundreds of moments, surpassing other off-the-shelf algorithms with a limit of a small number of moments. Based on this robust algorithm, we develop a new Maximum Entropy Method (MEMe) which improves upon the scalability of existing machine learning algorithms by efficiently approximating computational bottlenecks using maximum entropy and fast moment estimation techniques;We establish the link between maximum entropy methods and variational inference under moment constraints, hence connecting the former to well-known Bayesian approximation techniques;We apply MEMe to the problem of estimating the log determinant, crucial to inference in determinental point processes [11], and to that of estimating the entropy of a Gaussian mixture, important to state-of-the-art information-theoretic Bayesian optimisation algorithms.

## 2. Theoretical Framework

The method of maximum entropy, hereafter referred to as *MaxEnt* [12], is a procedure for generating the most conservative estimate of a probability distribution with the given information and the most non-committal one with regard to missing information [13]. Intuitively, in a bounded domain, the most conservative distribution, i.e., the distribution of maximum entropy, is the one that assigns equal probability to all the accessible states. Hence, the method of maximum entropy can be thought of as choosing the flattest, or most equiprobable distribution, satisfying the given moment constraints.

To determine the maximally entropic density q(x), we maximise the entropic functional
(1)$$S\;=\;-\;\int q\left(x\right)\log q\left(x\right)dx-\sum_{i=0}^{m}\alpha_{i}\left[\int q\left(x\right)x^{i}dx-\mu_{i}\right],$$
with respect to q(x), where the second term with $\mu_{i}=E_{p}\left[x^{i}\right]$ for some density p(x) are the power moment constraints on the density q(x), $\left\{\alpha_{i}\right\}$ are the corresponding Lagrange multipliers, and *m* is the number of moments. The first term in Equation (Equation 1) is referred to as the Boltzmann–Shannon–Gibbs (BSG) entropy, which has been applied in multiple fields, ranging from condensed matter physics [14] to finance [15]. For the case of i≤2, the Lagrange multipliers can be calculated analytically; for i≥3, they must be determined numerically.

In this section, we first establish links between the method of MaxEnt and Bayesian variational inference. We then describe fast moment estimation techniques. Finally, we present the proposed MaxEnt algorithm.

### 2.1. Maximum Entropy as Constrained Bayesian Variational Inference

The work of Bretthorst [16] makes the claim that the method of maximum entropy (MaxEnt) is fundamentally at odds with Bayesian inference. At the same time, variational inference [17] is a widely used approximation technique that falls under the category of Bayesian learning. In this section, we show that the method of maximum relative entropy [18] is equivalent to constrained variational inference, thus establishing the link between MaxEnt and Bayesian approximation.

#### 2.1.1. Variational Inference

Variational methods [10,17] in machine learning pose the problem of intractable density estimation from the application of Bayes’ rule as a functional optimisation problem:(2)$$p\left(z|x\right)=\frac{p(x|z)p(z)}{p(x)}\approx q\left(z\right),$$
where p(z) and p(z|x) represent the prior and posterior distributions of the random variable *z*, respectively. Variational inference therefore seeks to find q(z) as an approximation of the posterior p(z|x), which has the benefit of being a strict bound to the true posterior.

Typically, while the functional form of p(x|z) is known, calculating p(x)=∫p(x|z)p(z)dz is intractable. Using Jensen’s inequality, we can show that:(3)$$\log p\left(x\right)\geq E_{q}\left[\log p\left(x,z\right)\right]-E_{q}\left[\log q\left(z\right)\right].$$

Furthermore, the reverse Kullback-Leibler (KL) divergence between the posterior and the variational distribution, $D_{\mathrm{KL}}\left(q|p\right)$, can be written as:(4)$$\log p\left(x\right)=E_{q}\left[\log p\left(x,z\right)\right]-E_{q}\left[\log q\left(z\right)\right]+D_{\mathrm{KL}}\left(q|p\right).$$

Hence, maximising the evidence lower bound is equivalent to minimising the reverse KL divergence between *p* and *q*.

#### 2.1.2. MaxEnt Is Equivalent to Constrained Variational Inference

Minimising the reverse KL divergence between our posterior q(x) and our prior $q_{0}\left(x\right)$ as is done in variational inference, with respect to q(x):
(5)$$D_{\mathrm{KL}}\left(q|q_{0}\right)=-H\left(q\right)-\int_{x\in D}q\left(x\right)\log q_{0}\left(x\right)dx,$$
where H(q) denotes the differential entropy of the density q(x), such that ∫q(x)dx=1 and $\int q\left(x\right)x^{i}dx=\mu_{i}$. By the theory of Lagrangian duality, the convexity of the KL divergence, and the affine nature of the moment constraints, we maximise the dual form [19] of Equation (Equation 5):(6)$$\;-H\left(q\right)-\;\;\int \;\;q\left(x\right)\log q_{0}\left(x\right)dx-\;\sum_{i=0}^{m}\alpha_{i}\left(\int \;\;q\left(x\right)x^{i}dx-\mu_{i}\;\right)$$
with respect to q(x) or, alternatively, we minimise
(7)$$H\left(q\right)\;+\;\;\int \;q\left(x\right)\log q_{0}\left(x\right)dx+\;\sum_{i=0}^{m}\alpha_{i}\left(\int \;q\left(x\right)x^{i}dx-\mu_{i}\;\right).$$

In the field of information physics, the minimisation of Equation (Equation 7) is known as the method of relative entropy [18]. It can be derived as the unique functional satisfying the axioms of *locality*, *coordinate invariance*, *sub-system invariance* and *objectivity*.

The restriction to a functional is derived from considering the set of all distributions $Q=\{q_{j}\left(x\right)\}$ compatible with the constraints and devising a transitive ranking scheme (Transitive ranking means if A>B and B>C, then A>C). Furthermore, it can be shown that Newton’s laws, non-relativistic quantum mechanics, and Bayes’ rule can all be derived under this formalism [18]. In the case of a flat prior, we reduce to the method of maximum entropy with moment constraints [13] as $q_{0}\left(x\right)$ is a constant. Hence, MaxEnt and constrained variational inference are equivalent.

### 2.2. Fast Moment Estimation Techniques

As shown in Section 2.1, constrained variational inference requires the knowledge of the moments of the density being approximated to make inference. In this section, we discuss fast moment estimation techniques for both the general case and the case for which analytic expressions exist.

#### 2.2.1. Moments of a Gaussian Mixture

In some problems, there exist analytic expressions which can be used to compute the moments, One popular example is the Gaussian distribution (and the mixture of Gaussians). Specifically, for the one-dimensional Gaussian, we can find an analytic expression for the moments:(8)∫−∞∞xme−(x−μ)22σ2dx=∑i=0m(2σ)i+1miμm−iζi,
where μ and σ are the mean and standard deviation of the Gaussian, respectively, and:(9)$$\zeta_{i}=\int_{-\infty }^{\infty }y^{i}e^{-y^{2}}dy=\left\{\begin{array}{cc} \frac{\left(i-1\right)!!\sqrt{\pi }}{2^{i/2}}, & i\;\mathrm{even},\\ {}[1/2(i-1)!, & i\;\mathrm{odd}. \end{array}\right.$$
where (i−1)!! denotes double factorial. Hence, for a mixture of *k* Gaussians with mean $\mu_{k}$ and standard deviation $\sigma_{k}$, the *n*-th moment is analytic:(10)∑kwk∑i=0m(2σk)i+1miμkm−iζi.
where $w_{k}$ is the weight of *k*-th Gaussian in the mixture.

#### 2.2.2. Moments of the Eigenvalue Spectrum of a Matrix

Stochastic trace estimation is an effective way of cheaply estimating the moments of the eigenvalue spectrum of a given matrix $K\in R^{n\times n}$ [20]. The essential idea is that we can estimate the non-central moments of the spectrum by using matrix-vector multiplications.

Recall that for any multivariate random variable z with mean m and variance Σ, we have: (11)$$E\left(z^{T}z\right)=m^{T}m+\Sigma \overset{m=0,\Sigma =I}{=}I,$$
where for the second equality we have assumed (without loss of generality) that z possesses zero mean and unit variance. By the linearity of trace and expectation, for any number of moments m≥0 we have:(12)$$\sum_{s=1}^{n}\lambda_{s}^{m}=\mathrm{Tr}\left(IK^{m}\right)=E\left(zK^{m}z^{T}\right),$$
where $\left\{\lambda_{s}\right\}$ are the eigenvalues of *K*. In practice, we approximate the expectation in Equation (Equation 12) with a set of probing vectors and a simple Monte Carlo average, i.e., for *d* random unit vectors ${\left\{z_{j}\right\}}_{j=1}^{d}$:
(13)$$E\left(zK^{m}z^{T}\right)\approx \frac{1}{d}\left(\sum_{j=1}^{d}z_{j}K^{m}z_{j}^{T}\right),$$
where we take the product of the matrix *K* with the vector $z_{j}^{T}$ a total of *m* times and update $z_{j}^{T}$ each time, so as to avoid the costly matrix multiplication with $O(n^{3})$ complexity. This allows us to calculate the non-central moment expectations in $O(dmn^{2})$ complexity for dense matrices, or $O(dm\times n_{nz})$ complexity for sparse matrices, where d×m≪n and $n_{nz}$ is the number of non-zero entries in the matrix. The random vector $z_{j}$ can be drawn from any distribution, such as a Gaussian. The resulting stochastic trace estimation algorithm is outlined in Algorithm 1.

**Algorithm 1** Stochastic Trace Estimation [20]
1:**Input:** Matrix $K\in R^{n\times n}$, Number of Moments *m*, Number of Probe Vectors *d*2:**Output:** Moment Expectations $\hat{\mu_{i}}$, ∀i≤m3:Initialise $\hat{\mu_{i}}=0$
∀i4:
**for**
j=1,⋯,d
**do**
5:  Initialise $z_{j}=\mathrm{rand}\left(1,n\right)$6:  **for**
i=1,⋯,m
**do**
7:    $z_{j}^{T}=Kz_{j}^{T}$
8:    $\hat{\mu_{i}}=\hat{\mu_{i}}+\frac{1}{d}z_{j}z_{j}^{T}$
9:  **end for**
10:
**end for**



### 2.3. The Proposed MaxEnt Algorithm

In this section, we develop an algorithm for determining the maximum relative entropy density given moment constraints. Under the assumption of a flat prior in a bounded domain, this reduces to a generic MaxEnt density, which we use in various examples.

As we discussed previously, in order to obtain the MaxEnt distribution, we maximise the generic objective of Equation (Equation 1). In practice, we instead minimise the dual form of this objective [19], which can be written as follows:(14)$$S\left(q,q_{0}\right)=\int_{x\in D}q_{0}\left(x\right)\exp\left(-\left[1+\sum_{i=0}^{m}\alpha_{i}x^{i}\right]\right)dx+\sum_{i=0}^{m}\alpha_{i}\mu_{i}.$$

Notice that this dual objective admits analytic expressions for both the gradient and the Hessian:(15)$$\frac{\partial S(q,q_{0})}{\partial \alpha_{j}}=\mu_{j}-\int_{x\in D}q_{0}\left(x\right)x^{j}\exp\left(-\left[1+\sum_{i=0}^{m}\alpha_{i}x^{i}\right]\right)dx,$$
(16)$$\frac{\partial^{2}S\left(q,q_{0}\right)}{\partial \alpha_{j}\partial \alpha_{k}}=\int_{x\in D}q_{0}\left(x\right)x^{j+k}\exp\left(-\left[1+\sum_{i=0}^{m}\alpha_{i}x^{i}\right]\right)dx.$$

For a flat prior in a bounded domain, which we use as a prior for the spectral densities of large matrices and for Gaussian mixture models, $q_{0}\left(x\right)$ is a constant that can be dropped out. With the notation
(17)$$q_{\alpha }\left(x\right)=\exp\left(-\left[1+\sum_{i=0}^{m}\alpha_{i}x^{i}\right]\right),$$
we obtain the MaxEnt algorithm in Algorithm 2.

The input $\left\{\mu_{i}\right\}$ of Algorithm 2 are estimated using fast moment estimation techniques, as explained in Section 2.2.2. In our implementation, we use Python’s SciPy Newton-conjugate gradient (CG) algorithm to solve the minimisation in step 4 (line 6), having firstly computed the gradient within a tolerance level of ϵ as well as the Hessian. To make the Hessian better conditioned so as to achieve convergence, we symmetrise it and add jitter of intensity $\eta =10^{-8}$ along the diagonal. We estimate the given integrals with quadrature using a grid size of $10^{-4}$. Empirically, we observe that the algorithm is not overly sensitive to these choices. In particular, we find that, for well conditioned problems, the need for jitter is obviated and a smaller grid size works fine; in the case of worse conditioning, jitter helps improve convergence and the grid size becomes more important, where a smaller grid size leads to a computationally more intensive implementation.

**Algorithm 2** The Proposed MaxEnt Algorithm
1:**Input:** Moments $\left\{\mu_{i}\right\}$, Tolerance Level ϵ, Jitter variance in Hessian $\eta =10^{-8}$2:**Output:** Coefficients $\left\{\alpha_{i}\right\}$3:Initialise $\alpha_{i}=0$4:Compute gradient: $g_{j}=\mu_{j}-\int_{0}^{1}q_{\alpha }\left(x\right)x^{j}dx$5:Compute Hessian: $H_{jk}\;=\;\int_{0}^{1}q_{\alpha }\left(x\right)x^{j+k}dx,\;\;H\;=\;\frac{1}{2}\left(H+H^{T}\right)\;+\;\eta I$6:Minimize $\int_{0}^{1}q_{\alpha }\left(x\right)dx+\sum_{i}\alpha_{i}\mu_{i}$ using Conjugate Gradients until ∀j: $g_{j}<\epsilon$


Given that any polynomial sum can be written as $\sum_{i}\alpha_{i}x^{i}=\sum_{i}\beta_{i}f_{i}\left(x\right)$, where $f_{i}\left(x\right)$ denotes another polynomial basis, whether we choose to use power moments in our constraints or another polynomial basis, such as the Chebyshev or Legendre basis, does not change the entropy or solution of our objective. However, the performance of optimisers working on these different formulations may vary [21]. For simplicity, we have kept all the formulas in terms of power moments. However, we find vastly improved performance and conditioning when we switch to orthogonal polynomial bases (so that the errors between moment estimations are uncorrelated), as shown in Section 3.2.2. We implement both Chebyshev and Legendre moments in our Lagrangian and find similar performance for both.

## 3. Applications

We apply the proposed algorithm to two problems of interest, namely, log determinant estimation and Bayesian optimisation. In both cases we demonstrate substantial speed up and improvement in performance.

### 3.1. Log Determinant Estimation

Calculation of the log determinant is a common hindrance in machine learning, appearing in Gaussian graphical models [22], Gaussian processes [23], variational inference [24], metric and kernel learning [25], Markov random fields [26] and determinantal point processes [11]. For a large positive definite matrix $K\in R^{n\times n}$, the canonical solution involves the Cholesky decomposition of $K=LL^{T}$. The log determinant is then trivial to calculate as $\log\det\left(K\right)=2\sum_{i=1}^{n}\log L_{ii}$, where $L_{ii}$ is the ii-th entry of *L*. This computation invokes a computational complexity of $O(n^{3})$ and storage complexity of $O(n^{2})$, which becomes prohibitive for large *n*, i.e., $n>10^{4}$.

#### 3.1.1. Log Determinant Estimation as a Spectral Estimation Problem Using MaxEnt

Any symmetric positive definite (PD) matrix *K* is diagonalisable by a unitary matrix *U*, i.e., $K=U^{T}DU$, where *D* is the diagonal matrix of eigenvalues of *K*. Hence we can write the log determinant as:(18)$$\log\det\left(K\right)=\log\prod_{i}\lambda_{i}=\sum_{i=1}^{n}\log\lambda_{i}=nE_{p}\left(\log\lambda \right),$$
where we have used the cyclicity of the determinant and $E_{p}\left(\log\lambda \right)$ denotes the expectation under the spectral measure. The latter can be written as: (19)$$E_{p}\left(\log\lambda \right)=\int_{\lambda_{\min}}^{\lambda_{\max}}\;\;\;\;\;p\left(\lambda \right)\log\lambda d\lambda =\;\int_{\lambda_{\min}}^{\lambda_{\max}}\;\sum_{i=1}^{n}\frac{1}{n}\delta \left(\lambda -\lambda_{i}\right)\log\lambda d\lambda ,$$
where $\lambda_{\max}$ and $\lambda_{\min}$ correspond to the largest and smallest eigenvalues, respectively.

Given that the matrix is PD, we know that $\lambda_{\min}>0$ and we can divide the matrix by an upper bound of the eigenvalue, $\lambda_{\max}\leq \lambda_{u}$, via the Gershgorin circle theorem [27] such that:(20)$$\log\det\left(K\right)=nE_{p}\left(\log\lambda^{\prime }\right)+n\log\lambda_{u},$$
where $\lambda^{\prime }=\lambda /\lambda_{u}$ and $\lambda_{u}=\max_{i}(\sum_{j=1}^{n}|K_{ij}\left|\right)$, i.e., the maximum sum of the rows of the absolute of the matrix *K*. Hence we can comfortably work with the transformed measure:(21)$$\int_{\lambda_{\min}/\lambda_{u}}^{\lambda_{\max}/\lambda_{u}}p\left(\lambda^{\prime }\right)\log\lambda^{\prime }d\lambda^{\prime }=\int_{0}^{1}p\left(\lambda^{\prime }\right)\log\lambda^{\prime }d\lambda^{\prime },$$
where the spectral density $p(\lambda^{\prime })$ is 0 outside of its bounds of [0,1]. We therefore arrive at the following maximum entropy method (MEMe) for log determinant estimation detailed in Algorithm 3.

**Algorithm 3** MEMe for Log Determinant Estimation
1:**Input:** PD Symmetric Matrix $K\in R^{n\times n}$, Number of Moments *m*, Number of Probe Vectors *d*, Tolerance Level ϵ2:**Output:** Log Determinant Approximation logdet(K)3:
$\lambda_{u}=\max_{i}(\sum_{j=1}^{n}|K_{ij}\left|\right)$
4:
$B=K/\lambda_{u}$
5:Compute moments, $\left\{\mu_{i}\right\}$, via Stochastic Trace Estimation (Algorithm 1) with inputs (B,m,d)6:Compute coefficients, $\left\{\alpha_{i}\right\}$, via MaxEnt (Algorithm 2) with inputs $\left(\left\{\mu_{i}\right\},\epsilon \right)$7:Compute $q\left(\lambda \right)=\exp\left[-\left(1+\sum_{i}\alpha_{i}\lambda^{i}\right)\right]$8:Estimate $\log\det\left(K\right)\approx n\int \log\left(\lambda \right)q\left(\lambda \right)d\lambda +n\log\left(\lambda_{u}\right)$


#### 3.1.2. Experiments

#### Log Determinant Estimation for Synthetic Kernel Matrix

To specifically compare against commonly used Chebyshev [4] and Lanczos approximations [28] to the log determinant, and see how their accuracy degrades with worsening condition number, we generate a typical squared exponential kernel matrix [23], $K\in R^{n\times n}$, using the Python GPy package with 6 dimensional Gaussian inputs with a variety of realistic uniform length-scales. We then add noise of variance $\eta =10^{-8}$ along the diagonal of *K*.

We use m=30 moments and d=50 Hutchinson probe vectors [20] in MEMe for the log determinant estimation, and display the absolute relative estimation error for different approaches in Table 1. We see that, for low condition numbers, the benefit of framing the log determinant as an optimisation problem is marginal, whereas for large condition numbers, the benefit becomes substantial, with orders of magnitude better results than competing methods.

#### Log Determinant Estimation on Real Data

The work in Granziol and Roberts [29] has shown that the addition of an extra moment constraint cannot increase the entropy of the MaxEnt solution. For the problem of log determinant, this signifies that the entropy of the spectral approximation should decrease with the addition of every moment constraint. We implement the MaxEnt algorithm proposed in Bandyopadhyay et al. [2], which we refer to as OMxnt, in the same manner as applied for log determinant estimation in Fitzsimons et al. [30], and compare it against the proposed MEMe approach. Specifically, we show results on the Ecology dataset [31], with n=999,999, for which the true log determinant can be calculated. For OMxnt, we see that after the initial decrease, the error (Figure 1b) begins to increase for m>3 moments and the entropy (Figure 1a) increases at m=6 and m=12 moments. For the proposed MEMe algorithm, the performance is vastly improved in terms of estimation error (Figure 1d); furthermore, the error continually decreases with increasing number of moments, and the entropy (Figure 1c) decreases smoothly. This demonstrates the superiority both in terms of consistency and performance of our novel algorithm over established existing alternatives.

### 3.2. Bayesian Optimisation

Bayesian Optimisation (BO) is a powerful tool to tackle global optimisation challenges. It is particularly useful when the objective function is unknown, non-convex and very expensive to evaluate [32], and has been successfully applied in a wide range of applications including automatic machine learning [33,34,35,36], robotics [37,38] and experimental design [39]. When applying BO, we need to choose a statistical prior to model the objective function and define an acquisition function which trades off exploration and exploitation to recommend the next query location [40]. The generic BO algorithm is presented in Algorithm A1 in Appendix B. In the literature, one of the most popular prior models is the Gaussian processes (GPs), and the most recent class of acquisition functions that demonstrate state-of-the-art empirical performance is the information-theoretic ones [7,8,9,41].

#### 3.2.1. MaxEnt for Information-Theoretic Bayesian Optimisation

Information-theoretic BO methods select the next query point that maximises information gain about the unknown global optimiser/optimum. Despite their impressive performance, the computation of their acquisition functions involves an intractable term of Gaussian mixture entropy, as shown in Equation (Equation 22), if we perform a fully Bayesian treatment for the GP hyperparameters. Accurate approximation of this Gaussian mixture entropy term is crucial for the performance of information-theoretic BO, but can be difficult and/or expensive. In this paper, we propose an efficient approximation of the Gaussian mixture entropy by using MEMe, which allows for efficient computation of the information-theoretic acquisition functions.

As an concrete example, we consider the application of MEMe for Fast Information-Theoretic Bayesian Optimisation (FITBO) [9] as described in Algorithm 4, which is a highly practical information-based BO method proposed recently. The FITBO acquisition function has the following form:(22)$$\alpha \left(x|D_{t}\right)=H\left[\frac{1}{M}\sum_{j}^{M}p\left(y|I^{\left(j\right)}\right)\right]-\frac{1}{M}\sum_{j}^{M}H\left[p(y|I^{\left(j\right)})\right],$$
where $p\left(y|I^{\left(j\right)}\right)=p\left(y|D_{t},x,\theta^{\left(j\right)}\right)$ is the predictive posterior distribution for *y* conditioned on the observed data $D_{t}$, the test location x, and a hyperparameter sample $\theta^{\left(j\right)}$. The first entropy term is the entropy of a Gaussian mixture, where *M* is the number of Gaussian components.

The entropy of a Gaussian mixture does not have a closed-form solution and needs to be approximated. In FITBO, the authors approximate the quantity using numerical quadrature and moment matching (This corresponds to the maximum entropy solution for two moment constraints as well as the normalization constraint.). In this paper, we develop an effective analytic upper bound of the Gaussian mixture entropy using MEMe, which is derived from the non-negative relative entropy between the true density p(x) and the MaxEnt distribution q(x) [29]:(23)$$D_{\mathrm{KL}}\left(p||q\right)=-H\left(p\right)+H\left(q\right)\geq 0,$$
hence
(24)H(p)≤H(q).

Notice that q(x) shares the same moment constraints as p(x); furthermore, the entropy of the MaxEnt distribution q(x) can be derived analytically:(25)$$H\left(q\right)=1+\sum_{i=0}^{m}\alpha_{i}\mu_{i},$$
where $\left\{\mu_{i}\right\}$ are the moments of a Gaussian mixture, which can be computed analytically using Equation (Equation 10), and $\left\{\alpha_{i}\right\}$ are the Lagrange multipliers that can be obtained by applying Algorithm 2. The overall algorithm for approximating the Gaussian mixture entropy is then summarised in Algorithm 5. In Section B.1, we also prove that the moments of the Gaussian mixture uniquely determine its density, and the bound becomes tighter with every extra moment constraint: in the m→∞ limit, the entropy of the MaxEnt distribution converges to the true Gaussian mixture entropy. Hence, the use of a moment-based MaxEnt approximation can be justified [3].

**Algorithm 4** MEMe for FITBO
1:**Input:** Observed data $D_{t}$2:**Output:** Acquisition function $\alpha_{n}\left(x|D_{t}\right)$3:Sample *M* hyperparameters $\left\{\theta^{\left(j\right)}\right\}$4:
**for**
j=1,⋯,M
**do**
5: Approximately compute $p\left(y|D_{t},x,\theta^{\left(j\right)}\right)=N\left(y;m_{j},K_{j}\right)$ and its entropy $H\left[p(y|D_{t},x,\theta^{\left(j\right)})\right]$6:
**end for**
7:Approximate $H\left[\frac{1}{M}\sum_{j}^{M}N\left(y;m_{j},K_{j}\right)\right]$ with MEMe following Algorithm 58:Compute $\alpha (x|D_{t})$ as in Equation (Equation 22)


**Algorithm 5** MEMe for Approximating Gaussian Mixture Entropy
1:**Input:** A univariate Gaussian mixture $\mathrm{GM}=\frac{1}{M}\sum_{j}^{M}N\left(y;m_{j},\sigma_{j}^{2}\right)$ with mean $m_{j}$ and variance $\sigma_{j}^{2}$2:
**Output:**
$H_{\mathrm{GM}}\approx H\left[\frac{1}{M}\sum_{j}^{M}N\left(y;m_{j},\sigma_{j}^{2}\right)\right]$
3:Compute the moments of GM, $\left\{\mu_{i}\right\}$, analytically using Equation (Equation 10)4:Compute the Lagrange multipliers, $\left\{\alpha_{i}\right\}$, using Algorithm 25:
$H_{\mathrm{GM}}\approx -\left(1+\sum_{i}\alpha_{i}E\left[y^{i}\right]\right)=-\left(1+\sum_{i}\alpha_{i}\mu_{i}\right)$



#### 3.2.2. Experiments

#### Entropy of the Gaussian Mixture in Bayesian Optimisation

We first test a diverse set of methods to approximate the entropy of two sets of Gaussian mixtures, which are generated using FITBO with 200 hyperparameter samples and 400 hyperparameter samples on a 2D problem. Specifically, we compare the following approaches for entropy approximation: MaxEnt methods using 10 and 30 power moments (MEMe-10 and MEMe-30), MaxEnt methods using 10 and 30 Legendre moments (MEMeL-10 and MEMeL-30), variational upper bound (VUB) [42], method proposed in [43] with 2 Taylor expansion terms (Huber-2), Monte Carlo with 100 samples (MC-100), and simple moment matching (MM).

We evaluate the performance in terms of the approximation error, i.e., the relative error between the approximated entropy value and the true entropy value, the latter of which is estimated via expensive numerical integration. The results of the mean approximation errors by all methods over 80 different Gaussian mixtures are shown in Table 2 (The version with the standard deviation of errors is presented as Table A1 in Section B.2). We can see clearly that all MEMe approaches outperform other methods in terms of the approximation error. Among the MEMe approaches, the use of Legendre moments, which apply orthogonal Legendre polynomial bases, outperforms the use of simple power moments.

The mean runtime taken by all approximation methods over 80 different Gaussian mixtures are shown in Table 3 (The version with the standard deviation is presented as Table A2 in Section B.2). To ensure a fair comparison, all the methods are implemented in MATLAB and all the timing tests are performed on a 2.3 GHz Intel Core i5 processor. As we expect, the moment matching technique, which enables us to obtain an analytic approximation for the Gaussian mixture entropy, is the fastest method. MEMe approaches are significantly faster than the rest of approximation methods. This demonstrates that MEMe approaches are highly efficient in terms of both approximation accuracy and computational speed. Among all the MEMe approaches, we choose to apply MaxEnt with 10 Legendre moments in the BO for the next set of experiments, as it is able to achieve lower approximation error than MaxEnt with higher power moments while preserving the computational benefit of FITBO.

#### Information-Theoretic Bayesian Optimisation

We now test the effectiveness of MEMe for information-theoretic BO. We first illustrate the entropy approximation performance of MEMe using a 1D example. In Figure 2, the top plot shows the objective function we want to optimise (red dash line) and the posterior distribution of our surrogate model (blue solid line and shaded area). The bottom plot shows the acquisition functions computed based on Equation (Equation 22) using the same surrogate model but three different methods for Gaussian mixture entropy approximation, i.e., expensive numerical quadrature or Quad (red solid line), MM (black dash line), and MEMe using 10 Legendre moments (green dash line). In BO, the next query location is obtained by maximising the acquisition function, therefore the location instead of the magnitude of the modes of the acquisition function matters most. We can see from the bottom plot that MEMeL-10 results in an approximation that well matches the true acquisition function obtained by Quad. As a result, MEMeL-10 manages to recommend the same next query location as Quad. In comparison, the loose upper bound of the MM method, though successfully capturing the locations of the peak values, fails to correctly predict the global maximiser of the true acquisition function. MM therefore recommends a query location that is different from the optimal choice. As previously mentioned, the acquisition function in information-theoretic BO represents the information gain about the global optimum by querying at a new location. The sub-optimal choice of the next query location thus imposes a penalty on the optimisation performance as seen in Figure 3.

In the next set of experiments, we evaluate the optimisation performance of three versions of FITBO that use different approximation methods. Specifically, FITBO-Quad denotes the version that uses expensive numerical quadrature to approximate the entropy of the Gaussian mixture, FITBO-MM denotes the one using simple moment matching, and FITBO-MEMeL denotes the one using MEMe with 10 Legendre moments. We test these BO algorithms on two challenging optimisation problems, i.e., the Michalewicz-5D function [44] and the Hartmann-6D function [45], and measure the optimisation performance in terms of the immediate regret (IR): $\mathrm{IR}=|f^{*}-\hat{f}|$, which measures the absolute difference between the true global minimum value $f^{*}$ and the best guess of the global minimum value $\hat{f}$ by the BO algorithm. The average (median) result over 10 random initialisations for each experiment is shown in Figure 3. It is evident that the MEMe approach (FITBO-MEMeL-10), which better approximates the Gaussian mixture entropy, leads to a superior performance of the BO algorithm compared to the BO algorithm using simple moment matching technique (FITBO-MM).

## 4. Conclusions

In this paper, we established the equivalence between the method of maximum entropy and Bayesian variational inference under moment constraints, and proposed a novel maximum entropy algorithm (MEMe) that is stable and consistent for a large number of moments. We apply MEMe in two applications, i.e., log determinant estimation and Bayesian optimisation, to demonstrate its effectiveness and superiority over state-of-the-art approaches. The proposed algorithm can further benefit a wide range of large-scale machine learning applications where efficient approximation is of crucial importance.

## Figures and Tables

**Figure 1 entropy-21-00551-f001:**
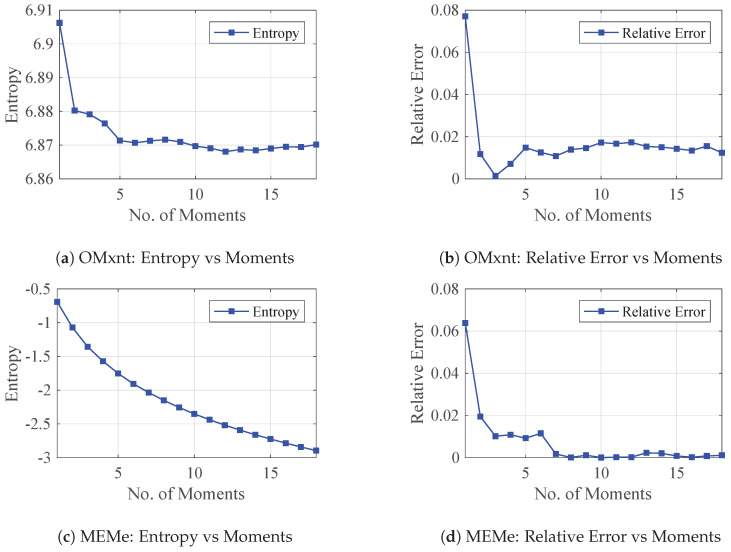
Comparison of the Classical (OMxnt) and our novel (MEMe) MaxEnt algorithms in log determinant estimation on real data. The entropy value (**a**) and estimation error (**b**) of OMxnt are shown in the top row. Those of the MEMe are shown in (**c**,**d**) in the bottom row.

**Figure 2 entropy-21-00551-f002:**
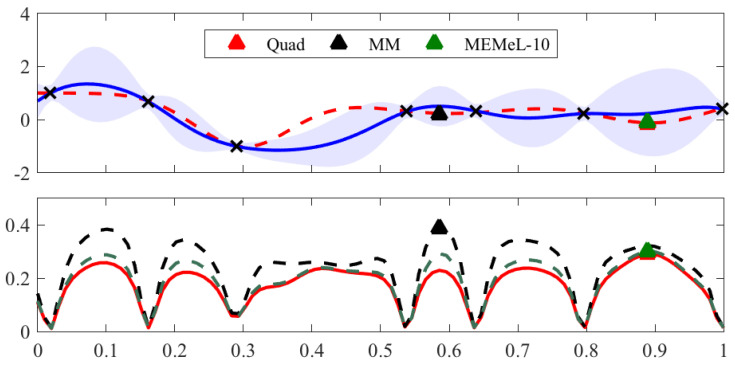
Bayesian Optimisation (BO) on a 1D toy example with acquisition functions computed by different approximation methods. In the top subplot, the red dash line is the unknown objective function, the black crosses are the observed data points, and the blue solid line and shaded area are the posterior mean and variance, respectively, of the GP surrogate that we use to model the latent objective function. The coloured triangles are the next query point recommended by the BO algorithms, which correspond to the maximiser of the acquisition functions in the bottom subplot. In the bottom plot, the red solid line, black dash line, and green dotted line are the acquisition functions computed by Quad, MM, and MEMe using 10 Legendre moments, respectively.

**Figure 3 entropy-21-00551-f003:**
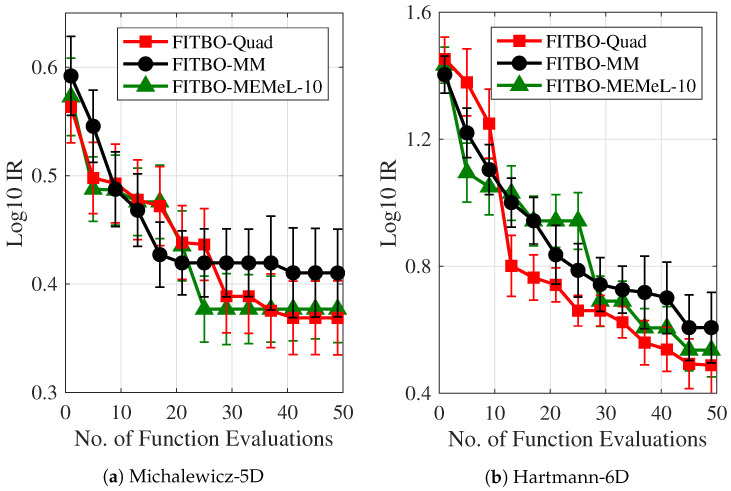
Performance of various versions of FITBO on 2 benchmark test problems: (**a**) Michalewicz-5D function and (**b**) Hartmann-6D function. The immediate regret (IR) on the *y*-axis is expressed in the logarithm to the base 10.

**Table 1 entropy-21-00551-t001:** Relative estimation error for MEMe, Chebyshev, and Lanczos approaches, with various length-scale *l* and condition number κ on the squared exponential kernel matrix $K\in R^{1000\times 1000}$.

κ	*l*	MEMe	Chebyshev	Lanczos
$3\times 10^{1}$	0.05	**0.0014**	0.0037	0.0024
$1.1\times 10^{3}$	0.15	0.0522	**0.0104**	0.0024
$1.0\times 10^{5}$	0.25	0.0387	0.0795	**0.0072**
$2.4\times 10^{6}$	0.35	0.0263	0.2302	**0.0196**
$8.3\times 10^{7}$	0.45	**0.0284**	0.3439	0.0502
$4.2\times 10^{8}$	0.55	0.0256	0.4089	0.0646
$4.3\times 10^{9}$	0.65	0.00048	0.5049	0.0838
$1.4\times 10^{10}$	0.75	**0.0086**	0.5049	0.1050
$4.2\times 10^{10}$	0.85	**0.0177**	0.5358	0.1199

**Table 2 entropy-21-00551-t002:** Mean fractional error in approximating the entropy of the mixture of *M* Gaussians using various methods.

Methods	M=200	M=400
MEMe-10	$1.24\times 10^{-2}$	$1.38\times 10^{-2}$
MEMe-30	$1.13\times 10^{-2}$	$1.06\times 10^{-2}$
MEMeL-10	$1.01\times 10^{-2}$	$0.85\times 10^{-2}$
MEMeL-30	$0.50\times 10^{-2}$	$0.36\times 10^{-2}$
VUB	$22.0\times 10^{-2}$	$29.1\times 10^{-2}$
Huber-2	$24.7\times 10^{-2}$	$35.5\times 10^{-2}$
MC-100	$1.60\times 10^{-2}$	$2.72\times 10^{-2}$
MM	$2.78\times 10^{-2}$	$3.22\times 10^{-2}$

**Table 3 entropy-21-00551-t003:** Mean runtime of approximating the entropy of the mixture of *M* Gaussians using various methods.

Methods	M=200	M=400
MEMe-10	$1.38\times 10^{-2}$	$1.48\times 10^{-2}$
MEMe-30	$2.59\times 10^{-2}$	$3.21\times 10^{-2}$
MEMeL-10	$1.70\times 10^{-2}$	$1.75\times 10^{-2}$
MEMeL-30	$4.18\times 10^{-2}$	$4.66\times 10^{-2}$
VUB	$12.9\times 10^{-2}$	$50.7\times 10^{-2}$
Huber-2	$20.9\times 10^{-2}$	$82.2\times 10^{-2}$
MC-100	$10.6\times 10^{-2}$	$40.1\times 10^{-2}$
MM	$2.71\times 10^{-5}$	$2.87\times 10^{-5}$

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
