# Peer review of "MEMe: An Accurate Maximum Entropy Method for Efficient Approximations in Large-Scale Machine Learning"

_entropy, 2019, doi:10.3390/e21060551_

Round 1
Reviewer 1 Report
This paper proposes an accurate maximum entropy method by first building the link between maximum entropy estimation and constrained variational inference and then optimizing the dual form of the entropy function. The technique is solid and the English is well-written. Some comments:
The authors did not write the detail of the proposed algorithm in text. It is hard for readers to follow the key of this method. It is necessary to present the detail clearly in the introduction section.
Maximum entropy approximation is very useful in large scale problems. Is it possible for authors to list more examples about where maximum entropy can be used? I suggest to expand the second paragraph in the introduction section
Eq.(14,15,16) i should be from 0 to m
Fig 2, 3, 4 is not clear in the print version (black and white), using different markers?
Authors mentioned that estimating the integral with quadrature is costly. Compared to quadrature, what is the time complexity advantage of the proposed method?
Author Response
We'd like to thank you for the constructive comments. The following are some of our responses:
Point 1: Is it possible for authors to list more examples about where maximum entropy can be used? I suggest to expand the second paragraph in the introduction section
Response 1: We've included the examples that we are aware of but will do more literature search in the near future. Also we would really appreciate if you could suggest some relevant literature.
Point 2: Authors mentioned that estimating the integral with quadrature is costly. Compared to quadrature, what is the time complexity advantage of the proposed method?
Response 2: Based on our empirical tests, the computational time of numerical quadrature is around 0.07 seconds in the case of M=400, which is twice that of our MEMe using 30 moments (0.03 seconds) and 4.7 times that of MEMe using 10 moments (0.015 seconds).
We’ve address all the other comments directly in the paper and all the changes are tracked or highlighted in the updated version. Again, we are grateful for your time and suggestions.
Reviewer 2 Report
Please see the attachment.

Author Response
We'd like to thank you for the constructive comments. The following are some of our responses:
Point 1: It is sometimes unclear what aspects of the paper are novel. For instance, is Section 2.2.2 a summary of a known technique from [20], or does it have some novelty?
The novel aspects of our paper are summarised in our contributions:
1) We propose a robust maximum entropy algorithm, which overcomes the limit of existing maximum entropy algorithms, and we use this to develop a novel Maximum Entropy Method (MEMe), which can provide computationally efficient approximations, e.g. for the two applications we identified: log determinant of a matrix and entropy of a Gaussian mixture.
2) We showcase the connection between maximum entropy methods and variational inference under moment constraints;
3) We empirically evaluate MEMe on the two applications: estimating the log determinant and approximating the entropy of a Gaussian mixture, and demonstrate the superior performance of MEMe over alternative methods.
The stochastic trace estimation is not part of our novel contribution but a useful tool we adopted for fast moment estimation. We’ve added the citation in the relevant algorithm to avoid confusion.
Point 2: Table 3: The baseline MM looks much faster. Is it possible that for certain fixed computational budgets, it could still be the choice with the best accuracy?
Moment-matching(MM) technique (i.e. fitting the Gaussian mixture with a single Gaussian) can give good estimation if the Gaussian mixture (GM) distribution resembles a single Gaussian distribution. Otherwise, our MEMe method which is able to harness more moment information (> 2 moments) of the GM distribution will lead to more accurate estimation. Moreover, our MEMe method, although slower than MM, is still very quick to compute as shown in Table 3 and we can speed it up further by reducing the number of moment constraints considered. In the special of using only 2 moment constraints, MEMe method is reduced to MM and enjoy the same speed advantage.
We’ve address all the other comments directly in the paper and all the changes are tracked or highlighted in the updated version. Again, we are very grateful for your time and helpful suggestions.